# Metabolic Pathway Engineering Improves Dendrobine Production in *Dendrobium catenatum*

**DOI:** 10.3390/ijms25010397

**Published:** 2023-12-28

**Authors:** Meili Zhao, Yanchang Zhao, Zhenyu Yang, Feng Ming, Jian Li, Demin Kong, Yu Wang, Peng Chen, Meina Wang, Zhicai Wang

**Affiliations:** 1Shenzhen Key Laboratory for Orchid Conservation and Utilization, The National Orchid Conservation Center of China and the Orchid Conservation & Research Center of Shenzhen, Shenzhen 518114, China; 17720798949@163.com (M.Z.); yangzhenyu.eco@foxmail.com (Z.Y.); lee799908@hotmail.com (J.L.); damykong@126.com (D.K.); 15361439572@163.com (Y.W.); chenpeng1003@163.com (P.C.); 2Key Laboratory of National Forestry and Grassland Administration for Orchid Conservation and Utilization, The National Orchid Conservation Center of China and the Orchid Conservation & Research Center of Shenzhen, Shenzhen 518114, China; 3State Key Laboratory for Conservation and Utilization of Subtropical Agro-Bioresources, South China Agricultural University, Guangzhou 510642, China; 13965002218@163.com; 4Shanghai Key Laboratory of Plant Molecular Sciences, College of Life Sciences, Shanghai Normal University, Shanghai 200234, China; fming@fudan.edu.cn

**Keywords:** *Dendrobium catenatum*, dendrobine, gene stacking, stress tolerance

## Abstract

The sesquiterpene alkaloid dendrobine, widely recognized as the main active compound and a quality control standard of medicinal orchids in the Chinese Pharmacopoeia, demonstrates diverse biological functions. In this study, we engineered *Dendrobium catenatum* as a chassis plant for the production of dendrobine through the screening and pyramiding of key biosynthesis genes. Initially, previously predicted upstream key genes in the methyl-D-erythritol 4-phosphate (MEP) pathway for dendrobine synthesis, including *4-(Cytidine 5′-Diphospho)-2-C-Methyl-d-Erythritol Kinase* (*CMK*), *1-Deoxy-d-Xylulose 5-Phosphate Reductoisomerase* (*DXR*), *2-C-Methyl-d-Erythritol 4-Phosphate Cytidylyltransferase* (*MCT*), and *Strictosidine Synthase 1* (*STR1*), and a few downstream post-modification genes, including *Cytochrome P450 94C1* (*CYP94C1*), *Branched-Chain-Amino-Acid Aminotransferase 2* (*BCAT2*), and *Methyltransferase-like Protein 23* (*METTL23*), were chosen due to their deduced roles in enhancing dendrobine production. The seven genes (*SG*) were then stacked and transiently expressed in the leaves of *D. catenatum*, resulting in a dendrobine yield that was two-fold higher compared to that of the empty vector control (*EV*). Further, RNA-seq analysis identified *Copper Methylamine Oxidase* (*CMEAO*) as a strong candidate with predicted functions in the post-modification processes of alkaloid biosynthesis. Overexpression of *CMEAO* increased dendrobine content by two-fold. Additionally, co-expression analysis of the differentially expressed genes (DEGs) by weighted gene co-expression network analysis (WGCNA) retrieved one regulatory transcription factor gene *MYB61*. Overexpression of *MYB61* increased dendrobine levels by more than two-fold in *D. catenatum*. In short, this work provides an efficient strategy and prospective candidates for the genetic engineering of *D. catenatum* to produce dendrobine, thereby improving its medicinal value.

## 1. Introduction

*Dendrobium catenatum*, an important traditional Chinese medicine and health food, is mainly distributed in the tropical and subtropical areas of southern China and some East Asian countries. Despite its popularity in China and longstanding use of over 2300 years, no cohort studies have been conducted on patients utilizing this plant. This rare and endangered species faces multiple threats in its native habitats. Hence, it is not advised to harvest from wild populations for health benefits [1]. To meet the challenges of harsh environments, *Dendrobium* plants accumulate high levels of bioactive ingredients, including polysaccharides, alkaloids, flavonoids, terpenes, and benzyl compounds [2], which are of high medicinal value. To meet the increasing demands for diverse healthy products, *D. catenatum* is commercially cultivated throughout China and some Asian countries, with an annual output value of more than 10 billion Chinese Yuan (equals to about USD 1.4 billion) [3,4]. However, cultivated *Dendrobium* was reported to produce inconsistent yields of compounds, largely influenced by the cultivar, harvesting season, plant age, and cultivation substrates [3,5]. In addition, the content of active compounds was also often found to vary strikingly among different tissues, with polysaccharides mainly found in stems, flavonoids and alkaloids in leaves, and bibenzyls in the roots and stems. The main bioactive compounds in *D. catenatum* include polysaccharides, polyphenols, terpenes, flavonoids, and alkaloids [1,6], of which alkaloids are one of the most important [7]. Despite the composition of alkaloids being extremely complex, 26 dendrobine-type sesquiterpenoid alkaloids (DSAs) were isolated and identified [8] due to their unique advantages in novel drug design and functional food development.

DSAs are based on the sesquiterpenoid skeleton of dendrobine, which was isolated from *D. nobile*, *D. findleyanum*, *D. linawianum*, *D. aurantiacum*, *D. snowflake*, *D. friedericksianum*, *D. signatum*, *D. moniliforme*, and *D. wardianum* [9]. Dendrobine has drawn much attention due to its broad application in health care, including therapeutic activities such as analgesic, antipyretic, antiviral, antihyperlipidaemic, antitumor, anti-Alzheimer’s disease, anti-inflammation, and blood pressure control [10]. Chemical synthesis of dendrobine reported earlier involved arduous steps with a low recovery rate [11]. Alternatively, dendrobine is mainly extracted from *Dendrobium* plants. However, due to the harsh environments and the slow growth rate of the *Dendrobium* species, dendrobine is only produced in very low amounts, even in *D. nobile*, hampering its supply to meet current industrial and research needs [12].

DSAs share a common sesquiterpene backbone primarily synthesized through two pathways, namely the mevalonate (MVA) pathway and the MEP pathway [13]. Transcriptome sequencing of *D. nobile* co-cultured with endophytic fungus *MF23* also identified 16 genes that were potentially involved in the synthesis of the sesquiterpene backbone [14]. Notably, genes encoding some enzymes, including Acetyl-CoA C-acetyltransferase (AACT), Phosphomevalonate Kinase (PMK), Mevalonate Diphosphate Decarboxylase (MVD), and post-modification enzymes including Cytochrome P450 1D10 (CYP1D10), METTL23, Histone-Lysine N-Methyltransferase 4 (ATX4), and BCAT2, were predicted to positively regulate dendrobine production, while *Terpene Synthase 21* (*TPS21*) showed a negative correlation to dendrobine accumulation [14]. However, due to the low levels of intermediates and the complex structure of dendrobine, the biosynthesis process of dendrobine and its key regulatory genes and enzymes involved were not studied extensively [15]. Therefore, it is crucial to explore new approaches for the synthesis of dendrobine, considering its high medicinal value.

Despite the total alkaloid content of *D. catenatum* being lower compared to that of *D. nobile*, the alkaloids in *D. catenatum* were of superior quality [16]. Similarly, while dendrobine content was detected high in *D. nobile*, it was only detected occasionally in *D. catenatum* [17]. In earlier studies, we sequenced the genome of *D. catenatum* [18], analyzed the biosynthetic pathways of dendrobine [19], identified alkaloid biosynthesis associated genes including *STR1* and *β-Glucosidases* (*BGLUs*), and verified *BGLU2*, *6*, *8*, *13* as regulators of alkaloid production [20,21]. Here, we pursued a novel synthetic biology approach for the high level production of dendrobine in *D. catenatum* leaves. Hub genes were screened, and the optimization of dendrobine biosynthesis using advanced gene stacking methods was performed. Additionally, the stacked genes were transformed into the genome of *A. thaliana* to verify their stress tolerance capacities. The metabolic engineering strategies used and the target genes characterized in this study were found to be beneficial for dendrobine biosynthesis, thus is promising in improving its yield in *Dendrobium* plants.

## 2. Results

### 2.1. Reconstitution of Multigene for Improved Dendrobine Synthesis

With little information being available on the cloning and characterization of genes in the deduced synthesis pathway of dendrobine [22], different combinations of the multigene were tried to optimize dendrobine production (Figure 1A). Considering that the precursor molecules of dendrobine could present in diverse plant species, it was possible to obtain dendrobine in *Dendrobium* plants or even from other plant species like *Nicotiana benthamiana* by introducing key genes in the synthesis pathway. The leaves of one-year-old *D. catenatum* seedlings were transiently infiltrated with various combinations of multigene constructs (*TG*, two genes; *FG*, five genes; *SG*, seven genes), where individual genes were earlier identified through transcriptome analysis by our research group [20] and other groups [23]. While the introduction of *TG* into *N. benthamiana* did not result in dendrobine production (suggesting the need for additional genes), enhanced production was achieved in *D. catenatum* leaves using the same constructs (Appendix A). Specifically, three upstream genes involved in precursor synthesis (*CMK*, *DXR*, and *MCT*), one node gene *STR1* encoding strictosidine synthase responsible for the conversion of tryptamine and secologanin into strictosidine, and three post-modification genes involved in oxidation/hydroxylation, transamination, and transmethylation processes (*CYP94C1*, *BCAT2*, and *METTL23*) were reconstituted (Figure 1B) and expressed in *D. catenatum* leaves using *Prrn* promotor and *TrbcL* terminator [24].

Leaf samples at 5 dpi were extracted and analyzed by high-performance liquid chromatography (HPLC). Dendrobine content significantly increased from 1.06 mg/g in the empty vector (*EV*) control to 1.20 mg/g in *STR1*-*CYP94C1* (*TG*) infiltrated leaves (Figure 1C), suggesting the efficient role of these two genes in the accumulation of dendrobine in *D. catenatum*. However, the five-gene combination (*FG*) did not affect dendrobine accumulation. Through a further combining as *SG*, the content of dendrobine dramatically elevated to 1.80 mg/g. The transcript abundance of each gene in the three multigene constructs expressed in *D. catenatum* leaves was checked by qRT-PCR analysis (Figure 1D). All the genes of interest were upregulated at varied levels except *MCT* in *FG*, which produced equal amounts of denbdrobine when compared to *EV*. *MCT* was highly expressed in *SG*, with the dendrobine content being remarkably elevated compared to *EV* and *TG*. Thus, *MCT* was identified as a key gene in regulating dendrobine synthesis.

To further confirm the potential role of *MCT* in dendrobine synthesis, it was transiently overexpressed in *D. catenatum* leaves, with the overexpression being verified using qRT-PCR (Figure 2A). Dendrobine content was then measured in the same samples, revealing a significant increase compared to the vector control (Figure 2B). To further determine the key function of *MCT* in dendrobine synthesis, CRISPR-dCas9 was utilized to decrease its expression level (Figure 2C). Consequently, dendrobine content showed a significant reduction (Figure 2D), confirming that *MCT* played a crucial role in enhancing dendrobine yield.

### 2.2. Overexpression of SG-Multigene Confers Salt and Drought Tolerance to Transgenic Arabidopsis

*D*. *catenatum* is a perennial epiphytic orchid that often faces abiotic stresses, such as salinity and drought [25]. Secondary metabolites play a significant role in helping plants to adapt to unfavorable environments [26]. Significant efforts were made to produce *SG*-multigene transgenic *D. catenatum*, but only a few plantlets were successfully regenerated (Figure 3A). The presence of transgenes was detected (Figure 3B) in *SG*-transgenic *D. catenatum* plantlets and the upregulation of *Farnesyl Diphosphate Synthase* (*FPPS*) as a marker gene for the activation of dendrobine synthesis pathway was recorded (Figure 3C). Despite a few plantlets being successfully transferred to pine-bark pots (Figure 3D,E), they were still too small to determine dendrobine content and stress response. As an alternative, the possible roles of *SG*-multigene in response to salt and drought stresses in *SG*-transgenic *Arabidopsis* were explored. The plants were obtained and characterized by PCR amplification of the resistance gene *hptII* (Figure 3F). To ensure no contamination, an analysis of the bacterial resistance gene *nptII* was also conducted. The expression levels of each gene in the *SG*-multigene cassette were analyzed as well (Figure 3G).

*Arabidopsis* seeds were germinated on a growth medium supplemented with different concentrations of NaCl (0, 100, 120, and 200 mM) or PEG6000 (0, 250, 400, 550 g/L) for varying periods (1, 3, and 4 weeks). *SG*-transgenic *Arabidopsis* exhibited an early flowering phenotype under normal conditions while showing increased salt tolerance compared to the vector control under stress conditions (Figure 4A). Specifically, *SG*-transgenic seedlings exposed to 100 mM NaCl treatment for four weeks gained more fresh weight (FW) (Figure 4B) in addition to the accumulation of lower levels of melondialdehyde (MDA) compared to the vector control transformants (Figure 4C). One-month-old plants exhibited a similar appearance and plant height (Figure 4D,E and Appendix A), indicating minimal influence of the transgene on plant development.

Similarly, *SG*-transgenic *Arabidopsis* showed greater resistance to drought stress induced by PEG6000 compared to the vector control (Figure 5A). After four weeks of treatment, transgenic plants gained more fresh weight (Figure 5B) and exhibited relatively lower MDA levels (Figure 5C) compared to the vector control. These results suggested that overexpression of *SG*-multigene could alter metabolite synthesis, thereby protecting plants from the damaging effects of environmental stress factors.

### 2.3. Identification of Downstream and Regulatory Genes Related to Dendrobine Synthesis

As genes associated with the downstream pathway of dendrobine synthesis are still largely unknown, further investigation was undertaken to identify those involved in downstream modification or regulatory processes. Alkaloids are important secondary metabolites mainly found in leaves and, to a lesser extent, in stems, but relatively rare in the roots of *Dendrobium* plants [27]. The read counts for a total of 17,327 genes from all 54 samples were normalized. Differentially expressed genes (DEGs) were identified in leaves and stems compared to roots and a Venn diagram was drawn to show the expression changes in the three *Dendrobium* plants. A total of 648 DEGs were detected in six comparison groups (Figure 6A, Appendix A). Further KEGG analysis of the 648 DEGs revealed that three of them could be associated with dendrobine biosynthesis (Figure 6B), with LOC110103175 (*PPOA1*, *Polyphenol Oxidase A1*) and LOC110109737 (*CMEAO*) being involved in isoquinoline alkaloid biosynthesis, and LOC110115472 (*SQMO*, *Squalene Monooxygenase-like*) involved in sesquiterpenoid and triterpenoid biosynthesis. Previous studies showed that inoculating *MF23*, a mycrorrhizal fungus obtained from the roots of *D. catenatum*, led to a significant increase (18.3%) in the content of dendrobine [23]. Additionally, it was found that the protocorm-like bodies (PLBs) of *D. catenatum* could accumulate higher amounts of alkaloids compared to the leaves [28]. These observations led us to investigate whether the associated transcriptomes were relevant to our findings. By combining Venn diagram analysis with the isolated 648 DEGs, a total of 41 overlapping DEGs (Figure 6C,Appendix A) were identified. Notably, one of the genes, *CMEAO*, predicted to be involved in the production of various alkaloids, including tropane, piperidine, pyridine, and isoquinoline alkaloids, reappeared (Figure 6D). Thus, it became highly likely that *CMEAO* played a crucial role in dendrobine biosynthesis.

### 2.4. Characterization of Novel Genes Associated with Dendrobine Accumulation

Genomic research on *Dendrobium* species provides abundant genetic resources, making it easier to correlate potential genes with important traits. WGCNA could be considered a principal method to screen candidate genes. To identify the key genes regulating dendrobine biosynthesis, gene co-expression analysis was conducted. In total, 648 DEGs were used as source data and subjected to the R package WGCNA. Six gene network modules belonging to six main branches of a hierarchical cluster tree were identified and represented in different colors (Figure 7A). A neighbor-joining heatmap was generated to visualize the interaction between these gene modules, showing strong correlations between them (Figure 7B,C). Subsequently, the correlation between gene expression in these modules and the content of alkaloid was evaluated (Figure 7D). Specifically, the module–trait relationship among 54 samples showed that the gene expression in the modules, turquoise vs. content/leaf (128 genes), yellow vs. content/leaf (53 genes), and grey vs. stem (only one gene, LOC110098898) were significantly correlated with the accumulation of alkaloids (Appendix A). To identify key genes, the top 20 hub genes based on gene significance (GS) as well as genes of interest in the turquoise, yellow, and grey modules were determined. The two genes of interest in the yellow vs. content/leaf module were LOC110094278 (*DLGT*, *Deoxyloganetic Acid Glucosyltransferase*) and LOC110104655 (*MYB61*), predicted to be involved in the synthesis of monoterpenoid indole alkaloids and in terpene metabolism in *Arabidopsis*, respectively. The two genes of interest in the turquoise vs. content/leaf module were LOC110098051 (*SAM-Mtase*, *S-Adenosyl-L-Methionine-dependent Methyltransferase*) and LOC110107874 (*CHLP*, *Geranylgeranyl Diphosphate Reductase*), associated with the flavonoid and several other metabolic pathways as well as terpenoid backbone biosynthesis in *Arabidopsis*, respectively. Lastly, LOC110098898 (*TLP1*, *Thaumatin-like Protein 1*) in the grey vs. stem module was the only enriched hub gene that was important for sugar metabolism in *Arabidopsis*.

Further, *CMEAO* and *MYB61* were evaluated for the regulation of dendrobine production. Each gene was transiently overexpressed in *D. catenatum* leaves, and the content of dendrobine in *D. catenatum* leaves at 5 dpi was determined through HPLC. Overexpression of *CMEAO*, identified through Venn diagram analysis (Figure 8A,B), and *MYB61*, isolated via WGCNA, significantly increased dendrobine content (Figure 8C,D). Specifically, overexpression of *CMEAO* resulted in a two-fold increase in dendrobine production, while overexpression of *MYB61* led to a more than two-fold increase. These findings clearly demonstrated that the previously isolated genes including *MCT*, *STR1*, *CYP94C1*, and the newly isolated *CMEAO* and *MYB61* played a crucial positive role in dendrobine metabolism, highlighting their potential as targets for enhancing dendrobine biosynthesis.

## 3. Discussion

Dendrobine, a sesquiterpene alkaloid with various bioactivities but limited resources, is regarded as a quality control for *D. nobile* [29]. However, in *D. catenatum*, a medicinal herb with growing commercial value, polysaccharides are abundant in the stems, with dendrobine being presented only in small amounts due to a complex and dynamic synthesis process governed by multiple factors [17]. Dendrobine levels vary depending on the plant’s development stages, source tissues, and local habitats. Despite genomic and transcriptomic studies identifying several genes related to the proposed dendrobine synthesis pathway, only a few were functionally confirmed [7]. Studies revealed that dendrobine might be necessary for young seedlings to defend against phytopathogenic microbes [13]. Similar situations occurred when the leaves of *D. catenatum* seedlings were challenged with *Agrobacterium*, leading to an increase in detectable levels of dendrobine. Further, increasing levels of dendrobine could enhance the medicinal quality and economic value of *D. catenatum*.

The sesquiterpene backbone of dendrobine is derived from the MVA and MEP pathways [16]. Sesquiterpene was detected and genes, including *DXR*, *CMK*, and *MCT* related to the MEP pathway, were annotated in *D. catenatum* [7,16]. The genes were predicted to be localized in the chloroplast (Appendix A). Moreover, *STR1*, initially identified as an essential enzyme involved in the terpenoid indole alkaloid (TIA) metabolism, and later deduced as a key player in dendrobine biosynthesis [22], was also identified in *D. catenatum* [16,20]. Here, overexpression of *STR1-CYP94C1* in the *TG*-multigene and co-expression with *MCT* in the *SG*-multigene significantly increased dendrobine production. It is important to note that the expression of these multigenes under the repetitive *Prrn* promoter may limit the full activation of these genes (Figure 1A). To address this limitation, alternative plastidal expressing promoters and plastid transformation vectors can be employed.

The growth and yield of *D. catenatum* are significantly influenced by environmental stresses, such as salinity and drought, which are common in wild cultivation [25]. One strategy resorted by plants to cope with these challenges is the accumulation of protective metabolic compounds, including flavonoids, sugars, and alkaloids [26]. In this study, the *SG*-multigene from *D. catenatum* was individually cloned, reconstituted, and introduced into *Arabidopsis*. Transgenic plants exhibited no noticeable morphological changes under normal growth conditions, except for early flowering. However, when exposed to salt and drought stresses, the *SG*-transgenic plants displayed improved tolerance, exhibiting enhanced growth and reduced cellular damage compared to the vector control, supporting earlier findings.

Alkaloids were one of the earliest identified bioactive compounds in *D. catenatum*. They present in multiple tissues with the highest content in PLBs [28], followed by a lesser content in leaves, and are very rare in roots [3]. In this study, WGCNA network analysis was performed based on the distribution of alkaloids and RNA-sequencing data from the four tissues of the four *Dendrobium* species. The present study aimed to identify the hub genes involved in alkaloid synthesis and to investigate their potential function in dendrobine synthesis. Among the selected genes, *CMEAO* and *MYB61* were verified as positive regulators for dendrobine production. *CMEAO* is a novel gene encoding copper methylamine oxidase and was predicted to be involved in tropane, piperidine, and pyridine alkaloid biosynthesis based on the KEGG analysis. The corresponding gene in *A. thaliana*, *Copper Amine Oxidase Zeta* (*CUAO-ZETA*, At2G42490), was found to share the highest similarity with *CMEAO*, whose function was not characterized yet. It was predicted to have copper binding and deaminating activities, which could be relevant in the post-modification processes of dendrobine biosynthesis. Overexpression of *CMEAO* in *D. catenatum* leaves resulted in a two-fold increase in dendrobine production, highlighting its significance in dendrobine synthesis. Increasing the expression levels of *CMEAO* could potentially enhance dendrobine production. Another gene, *MYB61*, was isolated in the yellow vs. content module and was predicted to be associated with terpene metabolism based on the KEGG analysis. Its homologue in *A. thaliana*, *AtMYB61*, was found to be involved in artemisinin biosynthesis [30]. It is worth noting that both artemisinin and dendrobine are sesquiterpenes derived from a common precursor, farnesyl diphosphate [11]. In this study, overexpression of *MYB61* from *D. catenatum* significantly increased dendrobine synthesis by more than two-fold, indicating its essential function in the process.

## 4. Materials and Methods

### 4.1. Plant Materials and Reagents

One-year-old *D. catenatum* plants were artificially cultivated in pine-bark pots and allowed to grow in a glass house at the Orchid Conservation & Research Center of Shenzhen (Shenzhen, China). The conditions provided were day/night temperature of 25/18 °C, relative humidity of 60%, and natural light. *N. benthamiana* and *A. thaliana* were grown on soil in a culture room on soil under a 16 h light/8 h dark photoperiod, with an illumination intensity of 2000 LX at 22 °C and a relative humidity of 65%. The dendrobine standard (CAS, 2115-91-5) was purchased from the National Institutes for Food and Drug Control (Beijing, China). Methanol, chloroform, and other regularly used chemicals (analytical grade) were purchased from Sangon Biotech Co., Ltd. (Shanghai, China).

### 4.2. WGCNA and Modular Characterization

Datasets for four *Dendrobium* species were obtained from the NCBI Sequence Read Archive (SRA) with the following accession numbers: SRP150489 and SRP274170 (*D. catenatum*), SRP122499 (*D. houshanense*), SRP139000 (*D. moniliforme*), and PRJNA338366 (*D. nobile*). A total of 54 samples were taken from four different tissues (stems, leaves, roots, and PLBs) of the four *Dendrobium* species. The raw reads were converted to fastq format, filtered using Trimmomatic, aligned to the reference genome of *D. catenatum* [18], and the gene expression levels were estimated as transcripts per million (TPM) using the Kallisto Super Wrapper V3 software package integrated in TBtools [31]. A |Log2 fold-change| > 1 threshold was used to identify DEGs in leaves vs. roots and stems vs. roots using the DESeq2 Wrapper in TBtools. DEGs were then analyzed using a Venn diagram (TBtools) and subjected to KEGG enrichment analysis (KOBAS). Gene co-expression networks were constructed using the R package WGCNA [32]. The network was built based on the scale-free topology criterion with a soft-thresholding power *β* of 14 (R^2^ > 0.87). The gene clustering tree was hierarchically established and cut using the dynamic shear tree algorithm, with random colors being assigned to each module for partitioning. A minimal module size of 30 and a branch merge cut height of 3.0 were used to identify the co-expression modules. The correlation of module eigengene values was calculated to isolate strong associations. Hub genes were selected based on the module weight value and correlation coefficient of module-alkaloid compounds. The top 20 hub genes with the highest correlation in each module, as well as genes of interest, were selected for further functional verification.

### 4.3. Vector Construction

Genes and primers used in this study are listed in Appendix A. The selected genes were amplified from the cDNA of *D. catenatum* using Phusion High-Fidelity DNA polymerase (New England Biolabs, Ipswich, MA, USA) and inserted into the pNC-Cam1304-*35S* vector [33]. Gene expression was regulated by the cauliflower mosaic virus 35S promoter and the *NOS* nopaline synthase terminator (NC Biotech, Haikou, China). The vector also included the *hptII* gene under the control of *35S* promoter and poly(A) terminator from the cauliflower mosaic virus. Multiple genes were stacked in a single plasmid of pYLTAC380H or pYL1300H using the unique nucleotide sequence-guided nicking endonuclease mediated DNA assembly (UNiEDA) [34]. Each gene in the multigene construct was assembled into a cassette containing the promotor of the rRNA operon from tobacco (*Prrn*), the leader sequence from *gene 10* of bacteriophage *T7* (*T7Lg10*), and the terminator of the chloroplast *rbcL* gene (*TrbcL*) [24]. Deactivated *Streptococcus pyogenes* Cas9 (d*Sp*Cas9) was introduced into the promotor region of *MCT* by sgRNA (5′-GATGCAAGAAAGAAAACCTA-3′) to knock down gene expression. The transient expression of dCas9 in *D. catenatum* leaves was performed following our previous protocol [21]. *Agrobacterium* EHA105 cells were transformed with plasmids containing the gene of interest via electroporation (2.5 kv, cuvettes with 0.1 cm gap). The correct clones were verified using diagnostic restriction digestion and Sanger sequencing (Beijing Genomics Institute, Beijing, China).

### 4.4. Genetic Transformation

*Agrobacterium* strain EHA105, harboring the genes of interest, was cultured at 28 °C in LB media supplemented with rifampicin (25 mg/L) and kanamycin (50 mg/L) until the OD_600_ of bacteria reached 0.6. The cultures were then centrifuged at 5000× *g* for 10 min, decanted, resuspended in a solution containing 10 mM 2-(N-morpholino) ethanesufonic acid, 10 mM MgCl_2_, and 200 μM acetosyringone to achieve a final OD_600_ of 1.5. The suspensions were incubated at room temperature for three hours. For transient expression, the suspensions were slowly infiltrated into the abaxial side of the leaves of one-year-old *D. catenatum* [21] or one-month-old *N. benthamiana* using a needleless syringe. Each group consisted of at least ten plants. After infiltration, the plants were allowed to grow under normal conditions for further analysis. *Agrobacterium* carrying the *EV* was used as the control. The infiltrated leaves were then harvested at various time points (6, 12, and 24 h for RNA extraction; 5 days for dendrobine measurement). The leaves were dried at 80 °C for three days and sent for dendrobine content determination using the Thermo Vanquish HPLC (Convinced-Test, Nanjing, China). For stable expression, the *Agrobacterium* suspensions were transformed into *A. thaliana* using the floral-dip method [35], or into *D. catenatum* using PLBs as explants [36]. Transgenic *Arabidopsis* seeds were harvested and selected on MS medium supplemented with 30.0 g/L sucrose, 7.8 g/L agar, and 25 mg/L hygromycin B. For salt treatment, *A. thaliana* seeds were germinated on MS medium supplemented with 30.0 g/L sucrose, 7.8 g/L agar, and different concentrations of NaCl (0, 100, 120, 200 mM), and grown for one month. For the drought treatment, MS medium supplemented with 25.0 g/L sucrose, 12.0 g/L agar, and different concentrations of PEG6000 (0.0, 250.0, 400.0, 550.0 g/L) was used. Plant phenotypes were monitored at various time points (1, 3, and 4 weeks). At the end of the treatment, fresh weight (each repeat containing 10 plants, n ≥ 5) and MDA content were measured (Convinced-Test, Nanjing, China). Non-treated seedlings (8-days-old) were transferred to pots (two seedlings per pot) with a mixture of vermiculite and organic soil (*v*/*v* = 1:3) and grown under normal culture conditions.

### 4.5. Metabolic Profiling

Transiently infiltrated tobacco leaves were allowed to grow for five days. Samples weighing 1.0 g each (n = 6) were collected and stored in liquid nitrogen. Extracts preparation, metabolic identification, and quantification were conducted at Novogene Biotechnology Co., Ltd. (Beijing, China), following standard procedures [37]. Principal component analysis (PCA) was used to detect outliers and evaluate batch effects. Supervised partial least square-discriminant analysis (PLS-DA) was used to elucidate the differences in metabolomes between groups. Metabolites with a relative importance cut-off value of 1.0 and |fold change| ≥ 2 were considered as differentially accumulated metabolites (DAMs). Finally, a heatmap based on the hierarchical cluster analysis method was generated using the R software (www.r-project.org).

### 4.6. qRT-PCR Validation of Gene Expression

A Quick RNA isolation Kit (cat. 0416-50, Huayueyang, Beijing, China) was used to isolate total RNA from frozen leaf samples following the manufacturer’s instructions. Reverse transcription was performed using a PrimeScript^TM^ RT reagent Kit with gDNA Eraser (cat. RR047B, Takara, Dalian, China). The resulting cDNA was diluted 10-fold and served as templates. qRT-PCR amplification was performed using Green qPCR MasterMix (cat. MT521-03, Biomed, Beijing, China) on an ABI PRISM 7500 Fluorescent Quantitative PCR System (Thermo Fisher Scientific, Singapore). *Actin7* gene (LOC104111011) of *D. catenatum* was used as the reference control. The primers used are listed in Appendix A. The comparative Ct method was used to calculate the relative gene expression with three biological replicates.

### 4.7. Data Analysis

All experiments were conducted as a randomized design. Data were expressed as means ± standard deviation (SD) and analyzed using GraphPad Prism 8 (La Jolla, CA, USA). An unpaired Student’s t-test was performed and *p* < 0.05 was considered statistically significant.

## Figures and Tables

**Figure 1 ijms-25-00397-f001:**
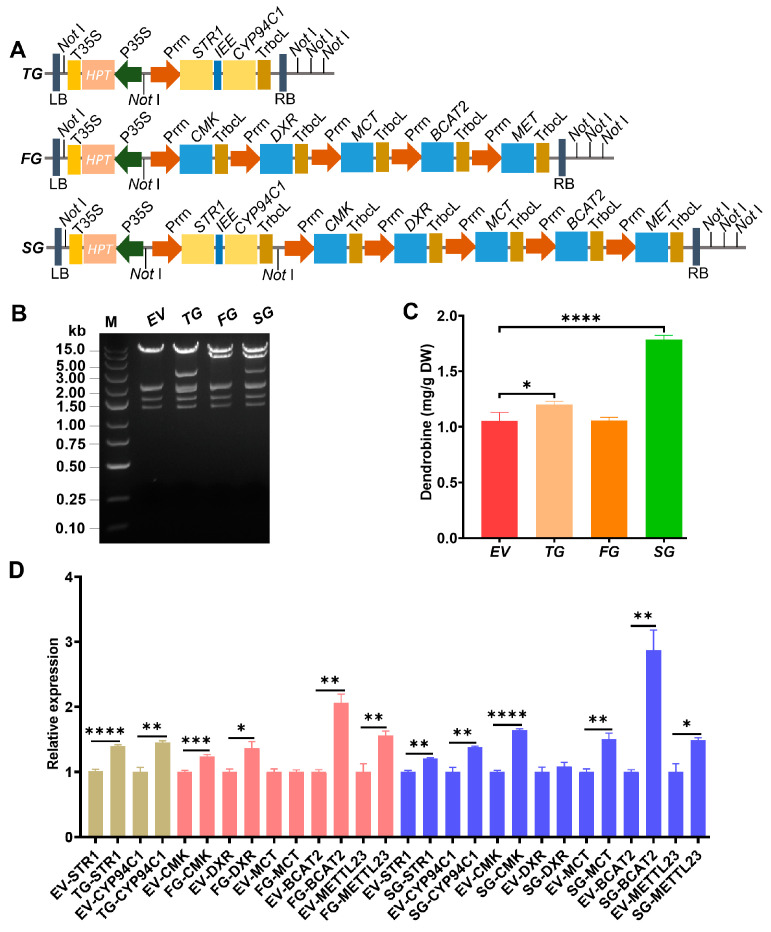
Multigene reconstruction to enhance dendrobine production. (**A**) Physical map of the multigene constructs used for integration and expression of the synthetic operons. The selected target genes are depicted as light blue boxes. Promotor *Prrn* is shown in dark yellow and terminator *TrbcL* in orange. The *HPTII* selectable marker gene for transformation is represented as a light yellow box. An intercistronic expression element (*IEE*) was put between *STR1* and *CYP94C1* operons to ensure the downstream cistron expression under the same promotor. (**B**) *Not* I-digestion analysis of pYLTAC380H-multigene (*EV*, *TG*, *FG*, and *SG*) constructs. M: DNA ladder marker. (**C**) Dendrobine content in *D. catenatum* leaves infiltrated with *Agrobacterium tumefaciens* carrying multigene constructs. There are three replicates for each sample. (**D**) The relative expression of each gene in a specific multigene construct. *EV*: empty vector; *TG*: two genes; *FG*: five genes; *SG*: seven genes. Asterisks indicate significance based on the Student’s *t*-test. * *p* ≤ 0.05; ** *p* ≤ 0.01; *** *p* ≤ 0.001; **** *p* ≤ 0.0001.

**Figure 2 ijms-25-00397-f002:**
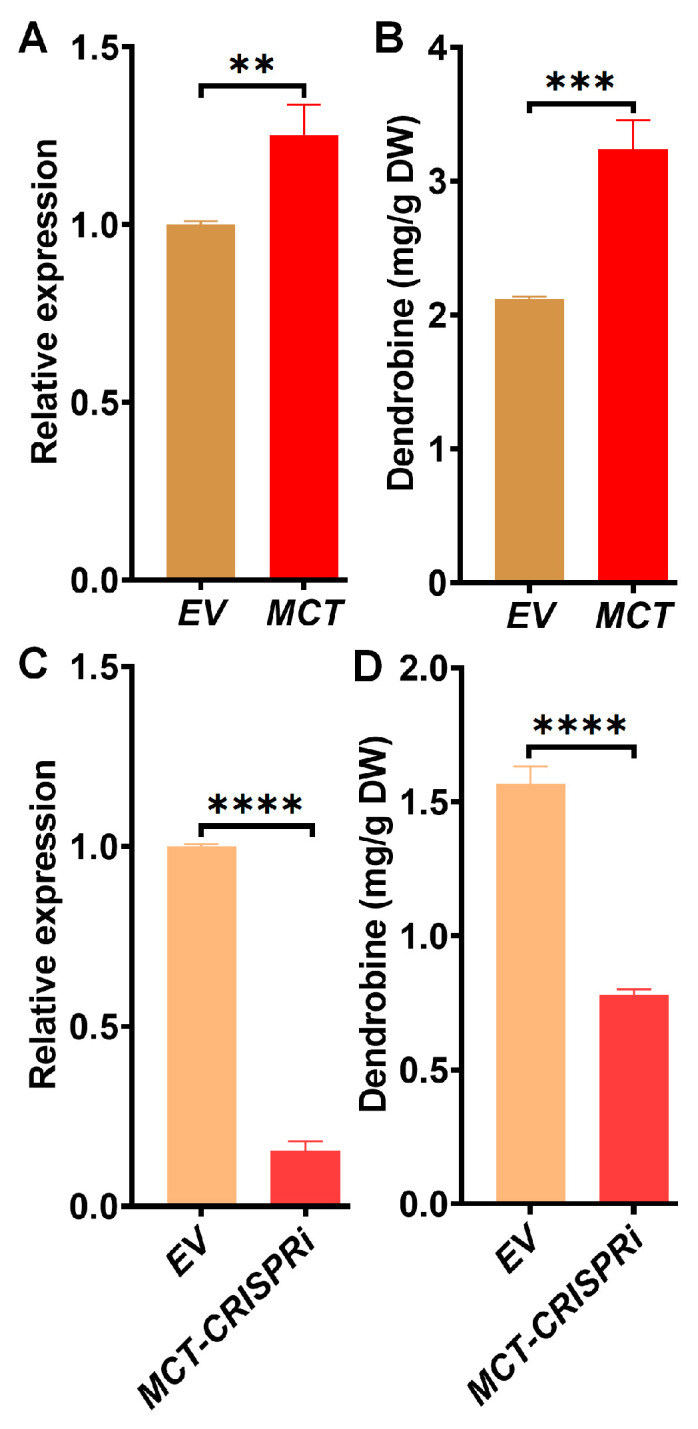
Functional verification of *MCT* in dendrobine synthesis. (**A**) *MCT* expression was checked by qRT-PCR with samples being collected 6 h after infiltration (n = 3). (**B**) Transiently infiltrated *D. catenatum* leaves were harvested (5 dpi) for dendrobine measurement (n = 3). (**C**) CRISPRi was performed to knock down *MCT* expression. The kinase-dead version of Cas9 (dCas9) was used to block transcription in the promotor region of *MCT*. Empty vector without dCas9 served as the control. *MCT* knock-down was verified by qRT-PCR, with samples being collected 6 h after infiltration (n = 3). (**D**) Leave samples were collected at 5 dpi and subjected to dendrobine measurement (n = 3). ** *p* ≤ 0.01; *** *p* ≤ 0.001; **** *p* ≤ 0.0001.

**Figure 3 ijms-25-00397-f003:**
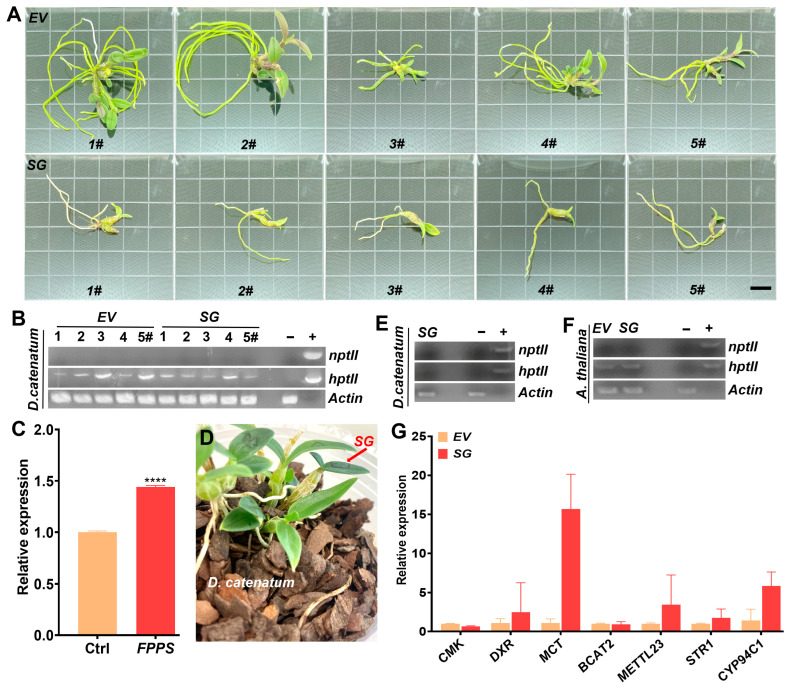
Generation and identification of multigene-transgenic plants. (**A**) Transgenic *D. catenatum* plantlets of *SG*-multigene (11-month-old) and *EV* control (13-month-old). (**B**) Analysis of the *hptII* transgene in *SG*-multigene and *EV* control transgenic *D. catematum* plantlets. DNA isolated from wild-type plants was used as the negative control “−”, while the *hptII* gene in the *EV* plasmid was used as the positive control “+”. *DcActin* was PCR amplified to demonstrate an equal amount of loading. *nptII* was amplified to avoid bacterial contamination. (**C**) *FPPS* expression was checked to demonstrate activation of the dendrobine synthesis pathway. *EV* transgenic *D. catenatum* served as the control (Ctrl). (**D**) *SG*-transgenic *D. catenatum* grown in pine-bark pots for 10 months. (**E**) Molecular characterization of the *hptII* transgene in *SG*-transgenic *D. catenatum* grown in pine-bark pots. (**F**) Molecular characterization of the *hptII* transgene in *SG*-transgenic *Arabidopsis*. (**G**) Expression analysis of individual genes in *SG*-transgenic *Arabidopsis* by qRT-PCR. Scale bar in (**A**) represents 1 cm. **** *p* ≤ 0.0001 represents significance.

**Figure 4 ijms-25-00397-f004:**
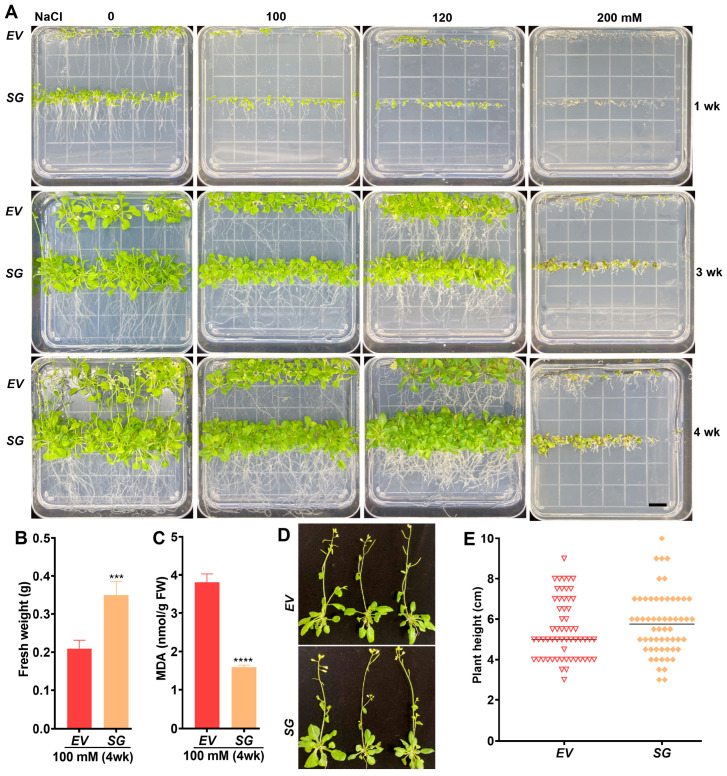
*SG*-transgenic *Arabidopsis* tolerant to salinity stress. (**A**) Plant growth in response to various concentrations of NaCl (0, 100, 120, and 200 mM). (**B**) Plant growth in terms of fresh weight under 100 mM NaCl for four weeks. (**C**) Cell damage in terms of MDA release under 100 mM NaCl for four weeks. (**D**) Representative image showing the transgenic plants growing in earth-pot for one month. (**E**) Comparison of plant height for the transgenic plants growing in earth-pot for one month. Data are represented as means ± SE from three replicates. *** *p* ≤ 0.001 and **** *p* ≤ 0.0001 are of significance compared to *EV* controls. Scale bar represents 1 cm. The center line in (**E**) represents median.

**Figure 5 ijms-25-00397-f005:**
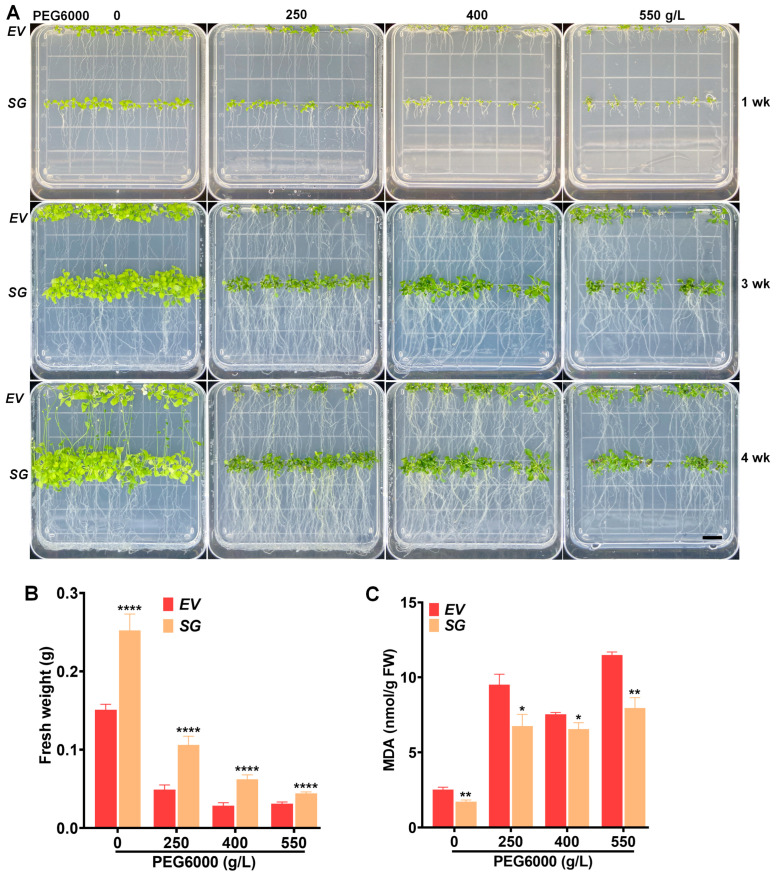
*SG*-transgenic *Arabidopsis* was tolerant to drought stress. (**A**) Plant growth in response to different concentrations of PEG6000 (0, 250, 400, and 550 g/L). (**B**) Plant growth in terms of fresh weight under varied concentrations of PEG6000 for four weeks. (**C**) Cell damage in terms of MDA release under varied concentrations of PEG6000 for four weeks. * *p* ≤ 0.05; ** *p* ≤ 0.01; **** *p* ≤ 0.0001 represent significance. Scale bar represents 1 cm.

**Figure 6 ijms-25-00397-f006:**
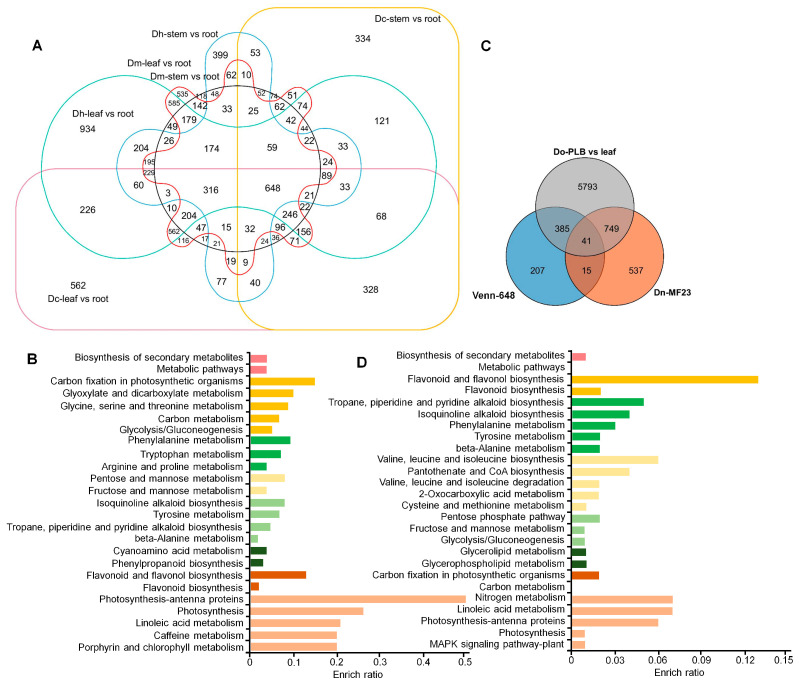
Dendrobine synthesis-related gene screening by Venn diagram and KEGG analysis. (**A**) Venn distribution of DEGs for transcriptomes of different tissues (stem vs. root; leaf vs. root) from three *Dendrobium* species (*D. houshanense*; *D. catenatum*, and *D. moniliforme*). (**B**) KEGG pathway enrichment of 648 DEGs from (**A**). The x-axis represents the enrichment ratio and the y-axis represents the pathway name. (**C**) Venn diagram representation of the number of DEGs in samples from protocorm-like bodies (PLBs), samples under *MF23* treatment, and the 648 DEGs in (**A**). (**D**) KEGG pathway enrichment of DEGs from (**C**).

**Figure 7 ijms-25-00397-f007:**
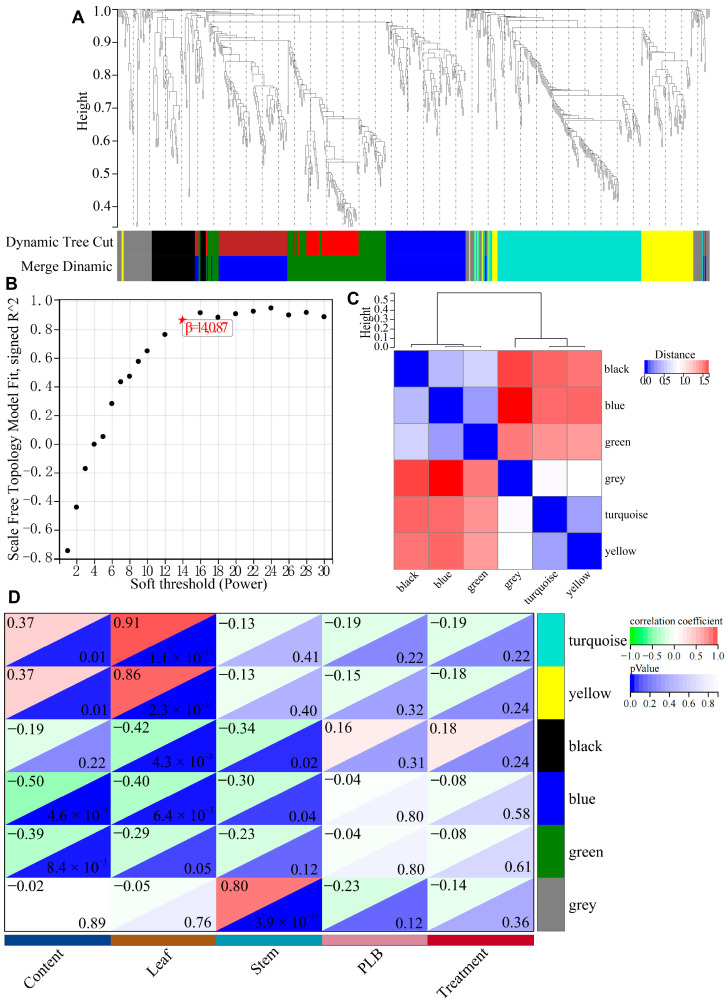
Dendrobine synthesis-related hub gene screening by WGCNA. (**A**) Hierarchical cluster dendrogram showing six expression modules of co-expressed genes. Each leaf in the tree represents an individual gene, with the branch representing a module of highly connected genes. The designated color rows below correspond to module membership. (**B**) Scale-free fit index at different threshold values (*β*). Asterisk indicates the selected soft-thresholding power. (**C**) Heatmap of connectivity of eigengenes. (**D**) Module-trait correlations and corresponding *p*-values (in parenthesis). The color in the box indicates −log(*P*) and the color scale indicates the *p*-value from the Fisher exact test. Treatment means *MF23* infection.

**Figure 8 ijms-25-00397-f008:**
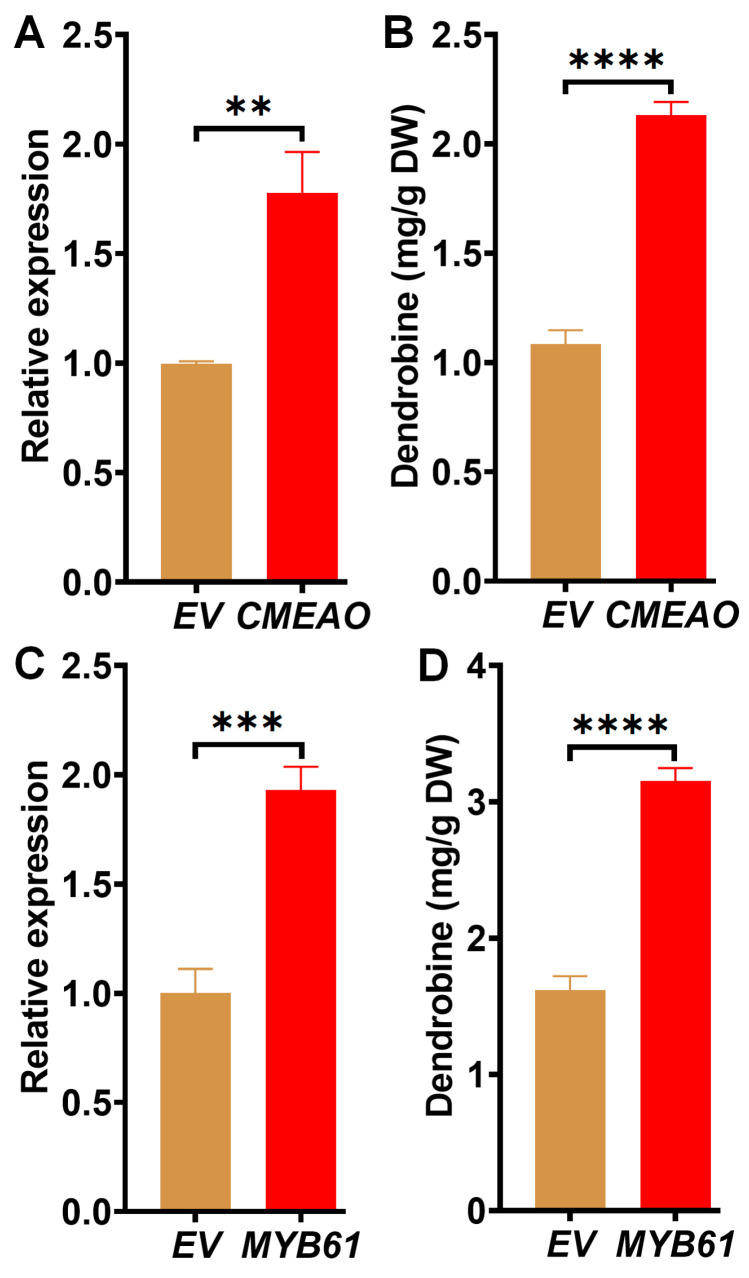
Functional verification of downstream genes in dendrobine synthesis. (**A**) *CMEAO* overexpression was verified by qRT-PCR. Samples were collected at 24 h post-infiltration. (**B**) Transiently infiltrated *D. catenatum* leaves (5 dpi) were harvested and subjected to dendrobine measurement. Empty vectors are used as controls. (**C**) qRT-PCR verification of *MYB61* overexpression in transiently infiltrated one-year-old *D. catenatum* leaves. (**D**) Dendrobine content in *D. catenatum* leaves transiently overexpressing *MYB61*. Statistical significance was demonstrated as following ** *p* ≤ 0.01, *** *p* ≤ 0.001, **** *p* ≤ 0.0001.

## Data Availability

The authors confirm that all data from this study are available and can be found in this article and in Appendix A.

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
