# Peer review of "Metabolic Pathway Engineering Improves Dendrobine Production in Dendrobium catenatum"

_ijms, 2023, doi:10.3390/ijms25010397_

Round 1

Reviewer 1 Report

Comments and Suggestions for Authors

The amount of work perfomed by the authors is impressive, and the significance of the results is high. I only have some minor questions.

Figure 4D: This panel is of very low quality and no difference can be observed. Can you prepare a picture with plants in the pot were diferences are easy to appreciate?

Introduction: I think is very specific and should be enhanced. Author mentions the use of Dendrobium as a medicinal plant, but there is any evidence ot it in humans? Are there any cohort studies in pacients using this plant? anu cochrane review. If there aren't, please mention it.

Line 59: can you put the equivalence in euros or dollars?

Author Response

Comments and Suggestions for Authors:

The amount of work performed by the authors is impressive, and the significance of the results is high. I only have some minor questions.

A: Thank you for your comments. Your input is greatly appreciated.

Figure 4D: This panel is of very low quality and no difference can be observed. Can you prepare a picture with plants in the pot were differences are easy to appreciate?

A: Thank you for your suggestions. We have replaced Figure 4D with a higher resolution photo. Furthermore, we have included representative images of earth-growing plants in Figure S2. As you can observe, the plants appear largely similar, and their heights are quite comparable (Figure 4E).

Introduction: I think is very specific and should be enhanced. Author mentions the use of Dendrobium as a medicinal plant, but is there any evidence using it in humans? Are there any cohort studies in patients using this plant? If there aren't, please mention it.

A: In Line45, we stated that Dendrobium catenatum is a significant traditional Chinese medicine and health food. Concerning cohort studies involving patients, we mentioned in Line47, “Despite its popularity in China and longstanding use of over 2300 years, no cohort studies have been conducted on patients utilizing this plant”.

Line 59: can you put the equivalence in euros or dollars?

A: We included the conversion calculation in Line56, stating “more than 10 billion Chinese Yuan (equivalent to approximately $ 1.4 billion).

Reviewer 2 Report

Comments and Suggestions for Authors

The paper "Metabolic Pathway Engineering Improves Dendrobine Production in Dendrobium catenatum" tackles the challenge of mass-producing dendrobine, a key medicinal compound in orchids. The study focuses on engineering D. catenatum by screening and pyramiding key biosynthesis genes, particularly those in the methyl-D-erythritol 4-phosphate (MEP) pathway. The authors successfully increased dendrobine yield two fold through transient and stable expression of selected genes. The investigation also identifies downstream genes like Copper Methylamine Oxidase (CMEAO) and the transcription factor MYB61, showcasing their potential in dendrobine biosynthesis. Overall, the paper provides an efficient strategy and promising genetic candidates for enhancing dendrobine production and elevating the medicinal value of D. catenatum. However, there is still room for imroving this work before it can be further processed.

Specific comments:

1. The abstract is too long and descriptive. It should be reduced by only presenting the most important findings of this work. 

2. My major concern is: What criteria were employed in selecting Dendrobium catenatum as the chassis plant for dendrobine production? Are there particular characteristics that make it well-suited for this purpose?

3. What evidence supports the selection of downstream post-modification genes (CYP94C1, BCAT2, METTL23) for their roles in enhancing dendrobine production? Were there alternative candidates considered?

4.  What is the rationale behind stacking and transiently expressing the seven selected genes in the chloroplast of D. catenatum leaves? How was the optimal combination determined?

5. The paper mentions increased tolerance to salt and drought stresses in transgenic Arabidopsis thaliana expressing the stacked genes. Can you provide insights into the mechanisms underlying this stress tolerance enhancement?

6. Why did BCAT2 and METTL23 in the downstream post-modification processes show little effect on dendrobine accumulation, especially in the context of the multigene infiltration (FG)?

7. Regarding the investigation to identify possible genes involved in downstream modification or regulatory processes, what specific methodologies were employed, and were any novel genes discovered?

8. Can you elaborate on the predicted functions of Copper Methylamine Oxidase (CMEAO) in post-modification processes? Were there any unexpected findings in its role in alkaloid biosynthesis?

9. Line 126-127: Could you elaborate on the choice of the plastid-specific promoter Prrn and terminator TrbcL for expressing the reconstituted genes in D. catenatum leaves?

10. Line 142-152: What factors influenced the differential expression of the genes of interest in the multigene constructs, especially the significant elevation of MCT in the SG construct?

11. Line 154-161: How was the decision made to focus on MCT for further investigation, and can you explain the methodology used in the transient overexpression and CRISPR-dCas9 experiments?

12. Line 172-176: What were the challenges faced in generating multigene-transgenic D. catenatum, and how did the upregulation of Farnesyl Diphosphate Synthase (FPPS) contribute to the activation of dendrobine synthesis pathway?

13. Line 201-206: Can you provide insights into the mechanisms underlying the early flowering phenotype observed in SG-transgenic Arabidopsis under normal conditions and the increased salt tolerance under stress conditions?

14. Line 234-249: What were the criteria for selecting PPOA1, CMEAO, and SQMO as potential genes associated with dendrobine biosynthesis, and how were these genes implicated in the alkaloid biosynthesis pathway?

15. Line 270-281: How were the hub genes identified, and what roles do DLGT, MYB61, SAM-Mtase, CHLP, and TLP1 play in terpene metabolism and other metabolic pathways in Arabidopsis?

Comments on the Quality of English Language

Moderate editing of English language required

Author Response

  1. The abstract is too long and descriptive. It should be reduced by only presenting the most important findings of this work.

A: Thank you for your suggestion. In order to be more concise, we have shortened the Abstract from 341 words to 241 words.

  1. My major concern is: What criteria were employed in selecting Dendrobium catenatum as the chassis plant for dendrobine production? Are there particular characteristics that make it well-suited for this purpose?

A: Among medicinal orchids, Dendrobium catenatum is a highly commercialized cultivar in China and some Asian countries (Line54). It is chosen as the plant chassis primarily for two reasons. The first reason, which is also the most significant, is the streamlined and standardized production process for commercial purposes. The second reason is its relatively short cultivation duration, ease of genetic transformation, and large biomass.

  1. What evidence supports the selection of downstream post-modification genes (CYP94C1, BCAT2, METTL23) for their roles in enhancing dendrobine production? Were there alternative candidates considered?

A: Li et al. (2017) (DOI:10.1038/s41598-017-00445-9) utilized transcriptome analysis to identify  genes associated with dendrobine synthesis. In their article, they described nine enzymes involved in post-modification. Specifically, we selected differentially expressed genes (DEGs) with a log2 fold change ≥ 2, including CYP94C1, BCAT2, and METTL23, and referenced their work (reference 23, Line117). Additionally, we attempted to incorporate other genes such as CYP71D55 into a plant-expression vector; however, we encountered difficulties in the construction process.

  1. What is the rationale behind stacking and transiently expressing the seven selected genes in the chloroplast of D. catenatumleaves? How was the optimal combination determined?

A: The expression of genes in the upstream pathway may enhance precursor production, while overexpression of genes in the downstream pathway genes may facilitate modification processes. Therefore, stacking genes from the upstream and downstream pathways might synergistically enhance dendrobine production (Line108~110). This strategy has been commonly used to modulate metabolite production, such as taxanes biosynthesis (Li et al, 2019, DOI: 10.1038/s41467-019-12879-y). Figure 1 clearly demonstrates that the stacking of TG and FG to obtain SG leads to significant increase in dendrobine accumulation. However, although this combination of SG is functional and shows promise in enhancing dendrobine production, it is not yet optimal, as indicated by our FG-multigene construct. Certain genes in the combination have minimal contribution, and some others may even have negative effect, which requires further elucidated in future research.

  1. The paper mentions increased tolerance to salt and drought stresses in transgenic Arabidopsis thalianaexpressing the stacked genes. Can you provide insights into the mechanisms underlying this stress tolerance enhancement?

A: In Line50-53, we depicted, “To meet the challenges of harsh environments, Dendrobium plants accumulate high levels of bioactive ingredients, including polysaccharides, alkaloids, flavonoids, terpenes, and benzyl compounds [2], which are of high medicinal value”. In Line352-354, “One strategy resorted by plants to cope with these challenges is the accumulation of protective metabolic compounds, including flavonoids, sugars, and alkaloids [26]”. In our study, the SG-multigene from D. catenatum was individually cloned, reconstituted, and introduced into Arabidopsis. The transgenic plants exhibited no noticeable morphological changes under normal growth conditions, except for early flowering. However, when exposed to salt and drought stresses, the SG-transgenic plants displayed improved tolerance, exhibiting enhanced growth and reduced cellular damage compared to the vector control (Line356-359). We propose that the enhanced stress tolerance might be attributed to the metabolic changes induced by the expressing of the SG-multigene.

  1. Why did BCAT2and METTL23in the downstream post-modification processes show little effect on dendrobine accumulation, especially in the context of the multigene infiltration (FG)?

A: BCAT2 and METTL23 has been implicated in post-modification processes of dendrobine production, while their specific roles in dendrobine synthesis have not been confirmed. In Figure 1D, it can be observed that the expression of BCAT2 in FG-multigene was upregulated by more than two-fold, while the expression of METTL23 showed a 1.5-fold increase. However, there was no noticeable change in dendrobine production in SG. This result indicates that the overexpression of these two genes has minimal impact on dendrobine production. It is possible that other regulatory factors, such as enzymatic activity, play a more significant role in dendrobine production.

  1. Regarding the investigation to identify possible genes involved in downstream modification or regulatory processes, what specific methodologies were employed, and were any novel genes discovered?

A: In Line239~243, “The read counts for a total of 17,327 genes from all 54 samples were normalized. Differentially expressed genes (DEGs) were identified in leaves and stems compared to roots and a Venn diagram was drawn to show the expression changes in the three Dendrobium plants. A total of 648 DEGs were detected in six comparison groups (Figure 6A)”. The gene list of these 648 DEGs can be found in Supporting File 1. Upon subjecting to KEGG analysis, only three candidates were identified as possibly being involved in dendrobine synthesis. These candidates are LOC110103175 (PPOA1, Polyphenol Oxidase A1) and LOC110109737 (CMEAO) in isoquinoline alkaloid biosynthesis, and LOC110115472 (SQMO, Squalene Monooxygenase-like) in sesquiterpenoid and triterpenoid biosynthesis (Figure 6B). None of these three genes have been previously characterized in relation to dendrobine synthesis.

  1. Can you elaborate on the predicted functions of Copper Methylamine Oxidase (CMEAO) in post-modification processes? Were there any unexpected findings in its role in alkaloid biosynthesis?

A: CMEAO is a novel gene that encodes copper methylamine oxidase. Based on the KEGG analysis, it was predicted to be involved in the biosynthesis of tropane, piperidine, and pyridine alkaloid biosynthesis (Line368). The corresponding gene in A. thaliana, Copper Amine Oxidase Zeta (CUAO-ZETA, At2G42490), showed the highest similarity with CMEAO, but its function has not been characterized yet. CMEAO is predicted to have copper binding and deaminating activities, which could be relevant in the post-modification processes of dendrobine biosynthesis (Line370~374). However, the exact chemical reactions mediated by CMEAO still requires further investigation. We also have supporting data indicating that overexpressing of CMEAO in D. nobile enhances dendrobine production, but this will be described in a separate publication.

  1. Line 126-127:Could you elaborate on the choice of the plastid-specific promoter Prrnand terminator TrbcL for expressing the reconstituted genes in D. catenatum leaves?

A: Prrn and terminator TrbcL are regulatory elements that are expected to be functional in plastid-containing tissues, such as stems and leaves (Occhialini et al., 2019,DOI: 10.1104/pp.18.01220), When expressed, proteins will be targeted to their respective destinations through signal peptides. We have predicted the cellular localization of proteins and summarized the results in Table S1. For example, MCT, DXR, and CMK were found to be located in the chloroplast, while MYB61 was found to be located in the nucleolus (Line341). It is important to note that the expression of these multigenes under the repetitive Prrn promoter may limit the full activation of these genes (Figure 1A). To address this limitation, alternative plastidal expressing promoters and plastid transformation vectors can be employed. This issue is further discussed in Line347-350.

  1. Line 142-152:What factors influenced the differential expression of the genes of interest in the multigene constructs, especially the significant elevation of MCTin the SG construct?

A: For unknown reasons, MCT has not been upregulated in FG, but it is expressed well in the SG-multigene construct. We speculate that the repetitive use of Prrn/TrbcL elements has somehow affected the full activation of certain individual genes in the multigene constructs.

  1. Line 154-161:How was the decision made to focus on MCTfor further investigation, and can you explain the methodology used in the transient overexpression and CRISPR-dCas9 experiments?

A: All the genes of interest were upregulated at varied levels except MCT in FG, which produced equal amounts of denbdrobine compared to EV. However, MCT was highly expressed in SG, with the dendrobine content being remarkably elevated compared to EV and TG. It is worth noting that CMK, DXR, BCAT2, and MET in the FG-multigene construct had minimal influence on dendrobine production, but they were still expressed in the SG-multigene construct. The expression of MCT in SG, along with STR1 and CYP94C1, appeared synergistically improve dendrobine production. Therefore, MCT was identified as a key gene in regulating dendrobine synthesis. In Line431-435, we described the CRISPR-dCas9 method used: “Deactivated Streptococcus pyogenes Cas9 (dSpCas9) was introduced into the promotor region of MCT by sgRNA (5'-GATGCAAGAAAGAAAACCTA-3') to knock down gene expression. Transient expression of dCas9 in D. catenatum leaves was performed following our previous protocol [21]”.

  1. Line 172-176:What were the challenges faced in generating multigene-transgenic D. catenatum, and how did the upregulation of Farnesyl Diphosphate Synthase(FPPS) contribute to the activation of dendrobine synthesis pathway?

A: Despite Agrobacterium tumefaciens-mediated genetic transformation efficiency being generally low in hormone-based tissue culture of orchids, several transformation and regeneration systems were developed in various orchid species. However, the method was time-consuming and inefficient, resulting in the production of only a small number of plants. Another challenge in the genetic transformation of orchids was the extended juvenile phase of immature tissues, hindering the functional analyses of genes. Additionally, compared to dicots, monocot orchids are more difficult to transform with A. tumefaciens as the natural hosts of the pathogen are dicots. It has been demonstrated that FPPS is closely correlated with dendrobine synthesis (Gong et al., 2022, DOI: 10.1186/s13568-022-01470-2). In our SG-multigene construct, MCT is responsible for producing precursors and is upstream of FPPS in the dendrobine synthesis pathway (Gong et al., 2021, DOI: 10.1007/s00253-021-11534-1). Therefore, we use upregulation of FPPS to indirectly indicate the activation of the dendrobine synthesis pathway (Figure 3C).

  1. Line 201-206:Can you provide insights into the mechanisms underlying the early flowering phenotype observed in SG-transgenic Arabidopsis under normal conditions and the increased salt tolerance under stress conditions?

A: In Line50~53, we mentioned, “To meet the challenges of harsh environments, Dendrobium plants accumulate high levels of bioactive ingredients, including polysaccharides, alkaloids, flavonoids, terpenes, and benzyl compounds [2], which are of high medicinal value”. It appears that the accumulation of these metabolites is an intelligent strategy employed by plants to respond to stresses. They need to expedite their development processes in order to complete their life cycle promptly. The expression of SG enhanced the production of metabolites but consumed more energy, preparing the plant for potential stress. This could possibly explain why Arabidopsis seedlings with high levels of metabolites exhibited a slightly earlier flowering phenotype. However, when subjected to salt and drought stresses, the SG-transgenic plants displayed improved tolerance compared to the vector control plant. They demonstrated enhanced growth and reduced cellular damage (Line356-360). We speculate that the improved stress tolerance is a result of metabolic changes induced by the expressing of the SG-multigene.

  1. Line 234-249:What were the criteria for selecting PPOA1, CMEAO, and SQMO as potential genes associated with dendrobine biosynthesis, and how were these genes implicated in the alkaloid biosynthesis pathway?

A: In Line239-243, “The read counts for a total of 17,327 genes from all 54 samples were normalized. Differentially expressed genes (DEGs) were identified in leaves and stems compared to roots and a Venn diagram was drawn to show the expression changes in the three Dendrobium plants. A total of 648 DEGs were detected in six comparison groups (Figure 6A)”. We provided the gene list of the 648 DEGs in Supporting File 1. When been subjected to KEGG analysis, only three candidates, namely LOC110103175 (PPOA1, Polyphenol Oxidase A1) and LOC110109737 (CMEAO) being involved in isoquinoline alkaloid biosynthesis, and LOC110115472 (SQMO, Squalene Monooxygenase-like) involved in sesquiterpenoid and triterpenoid biosynthesis are retrieved possibly related with alkaloid synthesis (Figure 6B). However, none of these three genes has been characterized before in dendrobine synthesis.

  1. Line 270-281: How were the hub genes identified, and what roles do DLGT, MYB61, SAM-Mtase, CHLP, and TLP1play in terpene metabolism and other metabolic pathways in Arabidopsis?

A: We described the WGCNA method for identifying hub genes in Line397-418. We utilized the R package WGCNA [32] to construct gene co-expression networks. We provided the gene lists retrieved from significantly correlated modules (Supporting File 3) . To identify key genes, we selected the top 20 hub genes based on gene significance (GS) and genes of interest in the turquoise, yellow, and grey modules. The two genes of interest in the yellow vs. content/leaf module were LOC110094278 (DLGT, Deoxyloganetic Acid Glucosyltransferase) and LOC110104655 (MYB61), predicted to be involved in the synthesis of monoterpenoid indole alkaloids and in terpene metabolism in Arabidopsis, respectively. The two genes of interest in the turquoise vs. content/leaf module were LOC110098051 (SAM-Mtase, S-Adenosyl-L-Methionine-dependent Methyltransferase) and LOC110107874 (CHLP, Geranylgeranyl Diphosphate Reductase), associated with the flavonoid and several other metabolic pathways as well as terpenoid backbone biosynthesis in Arabidopsis, respectively. Lastly, LOC110098898 (TLP1, Thaumatin-like Protein 1) in the grey vs. stem module was the only enriched hub gene that was important for sugar metabolism in Arabidopsis. It is worth mentioning that none of these genes have been characterized in dendrobine synthesis before.

Round 2

Reviewer 2 Report

Comments and Suggestions for Authors

The authors have addressed all concerns raised in the previous review, and the manuscript is well documented in the revised form and therefore i recommend the publication of this manuscript.